# Assessing the effects of disease-specific programs on health systems: An analysis of the Bangladesh Lymphatic Filariasis Elimination Program's impacts on health service coverage and catastrophic health expenditure

**Kimberly M. Koporc**[1]*, **David R. Hotchkiss**[1], **Charles F. Stoecker**[1], **Deborah A. McFarland**[2], **Thomas Carton**[1]

**1** Tulane University, School of Public Health and Tropical Medicine, New Orleans, Louisiana, United States of America, **2** Emory University, Rollins School of Public Health, Atlanta, Georgia, United States of America

☯ These authors contributed equally to this work.

* kkoporc@gmail.com

## Abstract

This study presents a methodology for using tracer indicators to measure the effects of disease-specific programs on national health systems. The methodology is then used to analyze the effects of Bangladesh's Lymphatic Filariasis Elimination Program, a disease-specific program, on the health system. Using difference-in-differences models and secondary data from population-based household surveys, this study compares changes over time in the utilization rates of eight essential health services and incidences of catastrophic health expenditures between individuals and households, respectively, of lymphatic filariasis hyper-endemic districts (treatment districts) and of hypo- and non-endemic districts (control districts). Utilization of all health services increased from year 2000 to year 2014 for the entire population but more so for the population living in treatment districts. However, when the services were analyzed individually, the difference-in-differences between the two populations was insignificant. Disadvantaged populations (i.e., populations that lived in rural areas, belonged to lower wealth quintiles, or did not attend school) were less likely to access essential health services. After five years of program interventions, households in control districts had a lower incidence of catastrophic health expenditures at several thresholds measured using total household expenditures and total non-food expenditures as denominators. Using essential health service coverage rates as outcome measures, the Lymphatic Filariasis Elimination Program cannot be said to have strengthened or weakened the health system. We can also say that there is a positive association between the Lymphatic Filariasis Elimination Program's interventions and lowered incidence of catastrophic health expenditures.

**Data Availability Statement:** The data underlying the results presented in this study are available from the Demographic and Health Survey Program (https://dhsprogram.com/) and the Bangladesh Bureau of Statistics (MD. Karamat Ali, Senior Programmer, Bangladesh Bureau of Statistics, Statistics and Informatics Division, mdkali0501@gmail.com, or Director General, Bangladesh Bureau of Statistics, dg@bbs.gov.bd, +88-02-5500-7056, bbs.gov.bd).

**Funding:** The authors received no specific funding for this work.

**Competing interests:** The authors have declared that no competing interests exist.

## Author summary

Evidence to understand the interactions between disease specific programs and the health system is insufficient and largely based on opinion. This study presents a methodology for using tracer indicators to measure the effect of a disease-specific program, the Bangladesh Lymphatic Filariasis Elimination Program, on its health system. The Composite Coverage Index and incidence of catastrophic health expenditures are well-established tracer indicators for measuring the strength of a health system. In this study, they were calculated, before the program started in 2000 and after it ended in 2015, using data from Demographic and Health Surveys and Household Income and Expenditure Surveys, respectively. Using the Composite Coverage Index to measure the effects of the Lymphatic Filariasis Elimination Program revealed that it did not negatively or positively affect health service coverage rates. We can also say that there is a positive association between the program interventions and lowered incidence of catastrophic health expenditures.

## Introduction

The definition of Universal Health Care (UHC) is "a desired outcome of health system performance, whereby all people who need health services (promotion, prevention, treatment, rehabilitation, and palliation) receive them without incurring financial hardship" [1]. UHC has been described as the goal of health systems strengthening [2]. A health system can be strengthened by increasing inputs and using appropriate strategies to enhance its functions, which are service provision, governance, financing, health workforce development, health information systems management, and supply chain management [3].

Disease-specific programs (DSP), including neglected tropical disease (NTD) programs, have been criticized for their potential to distort and overburden health systems [4,5]. Cavalli et al. conducted a district-level cross-sectional study to document positive and negative effects on Mali's health system. The most concerning result of the study was that the increased workload from mass drug administrations (MDAs) interfered with or interrupted routine care. The study also brought attention to important issues regarding DSPs integration into the existing health system, such as the need to: involve district health management teams in decision making; ensure training is relevant and adequate; integrate parallel systems that develop as a result of MDAs (e.g., workforce, drug supply chains, and health management information systems); and limit unnecessary bureaucracy [6]. Other issues of concern include possible increases in inequity because services may be more directed toward accessible populations to meet targets, and the undermining of government authority due to siloed DSPs with conflicting objectives [7,8].

Proponents of DSPs argue that the programs can be integrated in a way that provides maximum benefits to the health system [9–14]. However, according to the World Health Organization (WHO), evidence to understand the interactions between DSPs and the health system is insufficient, and the existence of both positive and negative associations suggests that the way they interact is important [7,8].

This study contributes to the evidence by presenting a methodology for measuring the effects of DSPs on national health systems. The methodology is then used to analyze the effects of Bangladesh's Lymphatic Filariasis Elimination Program (LFEP), an NTD program and a DSP, on the health system. This study uses difference-in-differences (DID) models and secondary data from Demographic and Health Surveys (DHS) and Household Income and

**Table 1. WHO and WB's list of tracer indicators for measuring progress toward UHC.**

| Coverage Indicators | | Equity in Coverage | Financial Protection |
|---|---|---|---|
| **Promotion and Prevention Services** | **Treatment Services** | | |
| • Family planning coverage with modern methods<br>• Antenatal care coverage<br>• Skilled birth attendance<br>• Diphtheria, tetanus and pertussis immunization coverage among 1-year-olds<br>• Prevalence of no tobacco smoking in the past 30 days among adults ≥ 15 years<br>• % of population using improved drinking water sources<br>• % of population using improved sanitation facilities<br>• Preventive chemotherapy coverage against NTDs | • Antiretroviral treatment<br>• Tuberculosis treatment<br>• Hypertension coverage<br>• Diabetes coverage<br>• Cataract surgical coverage | • Wealth quintiles<br>• Location of residence (rural or urban)<br>• Gender | • Percentage of population not spending more than 25% of non-food expenditure<br>• Percentage of the population neither impoverished by out-of-pocket payments nor pushed further into poverty by them |

Expenditure Surveys (HIES) to measure two components of UHC, which are key goals of health systems strengthening: access to essential healthcare services; and financial protection against catastrophic health expenditures (CHE)[1,2].

In 2014, drawing upon indicators established for monitoring the Millennium Development Goals, WHO and the World Bank (WB) established a list of tracer indicators to measure progress toward UHC. The list included a core set of indicators for measuring coverage with essential services, equity in coverage (e.g., gender and wealth quintiles), and financial protection (e.g., the incidence of CHE and the incidence of impoverishment due to household out of pocket medical payments (OOP)). The set of tracer indicators was updated in 2015 to include indicators for noncommunicable diseases and diseases of the poor (e.g., NTDs) [15,16]. The list of tracer indicators is presented in Table 1.

To understand the appropriateness of and relationships among the indicators studied, the conceptual model (Fig 1) was adapted from the logic model developed by Boerma et al. [1]. It

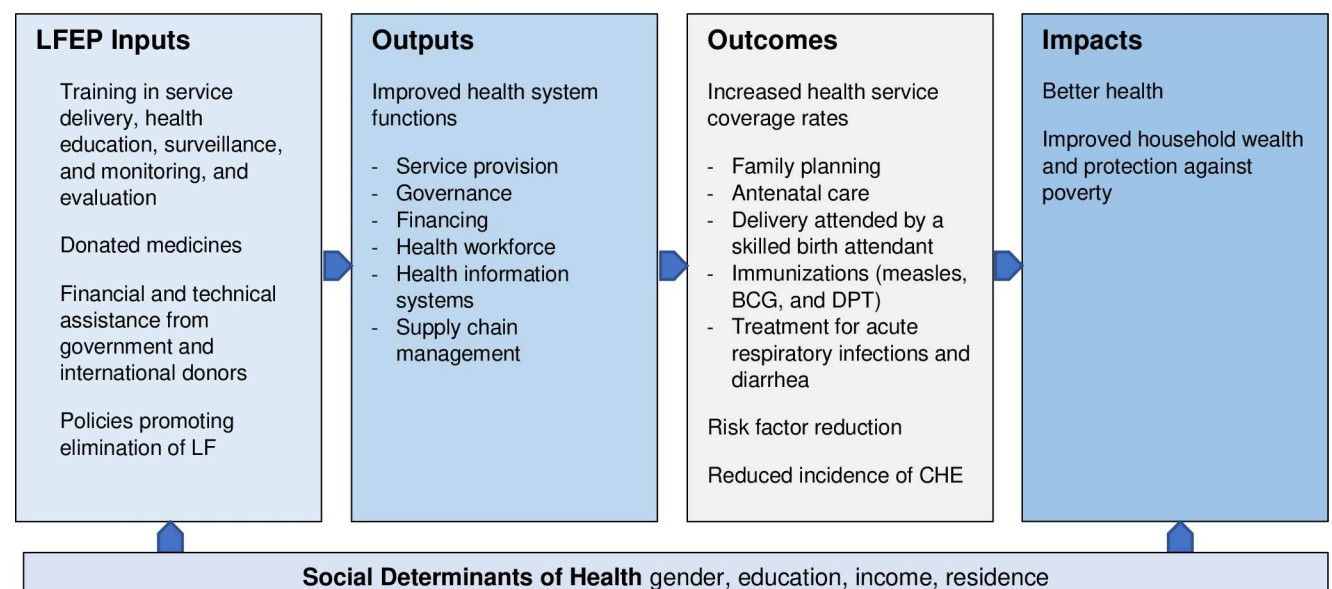

**Fig 1. Conceptual Model–Logistic model for monitoring health system functions.**

illustrates how program inputs affect health system outputs (i.e., quality of health system functions), which, in turn, affect the health system's ability to generate outcomes (i.e., provide health services). Ultimately, the health system outcomes have an impact on poverty and health [1]. The inputs into the system include workforce training, donated medicines, and financial and technical assistance, which strengthen functions and other services.

Regarding inputs of Bangladesh's LFEP, albendazole and diethylcarbamazine citrate for the MDA are donated by GalxoSmithKline and Eisai, respectively, for the MDA. The donation is administered and monitored by WHO. In the context of NTDs, an MDA is the distribution of medicine to 100% of the targeted population, once or twice a year, irrespective of an individual's disease status. Pregnant women, children aged less than two years, and the severely ill are excluded from participating in MDAs. The drugs are distributed by trained community health workers and teachers at the community level with oversight by ministry of health staff [17]. Other interventions include morbidity management and disability prevention (MMDP), vector control, reduction of environmental risk factors, disease surveillance, monitoring and evaluation (M&E) and information, education, and communication (IEC) activities [18].

The district is the implementation unit, which means the entire targeted population within the district is treated if there is documented evidence of lymphatic filariasis (LF) at a specified prevalence or level of intensity anywhere in the district. Districts are considered hyper-endemic when the average resident population, or any subunit of the population, has an antigenemia or a microfilaremia positivity rate equal to or greater than 1% [19].

The outcomes measured are the Composite Coverage Index (CCI), each essential health service tracer indicator (e.g., family planning, antenatal care, and immunization) that makes up the index [20–23], and CHE incidence. These health services and CHE incidence are not related to the LFEP, but they are used in this analysis because they are the tracer indicators established by the WHO and the WB to measure progress toward UHC–an outcome of a strong health system.

## Methodology

This study measures the changes in health service coverage rates and CHE incidence in treatment and control districts overtime and compares the differences in changes between the two groups while controlling for social determinants of health (e.g., level of education, socio-economic status, and residence (i.e., urban vs rural)).

Of Bangladesh's 64 districts, 19 were LF hyper-endemic and targeted by the LFEP. MDAs were implemented for at least five consecutive years between 2001 to 2015. By 2015, all 19 hyper-LF-endemic districts were declared LF free, and the MDAs were stopped. Coverage rates for each MDA are presented in Table 2. Out of a total of 146 MDAs implemented over the 15-year period, 14 (<10%) had no treatment data. Of the remaining 132 MDAs for which there were treatment data, only two (<2%) did not reach the desired treatment WHO target of 65%. Other activities implemented by the LFEP include IEC, MMDP, M&E, and surveillance [18,24].

The other 45 districts were not targeted because they were either hypo- or non-endemic (Shamsuzzaman et al., 2017). They are the control districts. The independent variable of interest is whether the district was targeted by the LFEP. It is a binary variable (yes/no).

## Data

The CCI and health service coverage rates are calculated using DHS data provided by the DHS program. CHEs are calculated using HIES data provided by the Bangladesh Bureau of Statistics.

**Table 2. MDA treatment coverage rates (%) by year (2001–16) and LF-endemic district.**

| LF-Endemic District | '01 | '02 | '03 | '04 | '05 | '06 | '07 | '08 | '09 | '10 | '11 | '12 | '13 | '14 | '15 | '16 | Total MDAs |
|---|---|---|---|---|---|---|---|---|---|---|---|---|---|---|---|---|---|
| **Barisal Division** | | | | | | | | | | | | | | | | | |
| Barguna | | | | | 89* | 90 | ND | 91* | 95* | 95* | S | | | | | | 6 |
| Barisal | | | | | | | | 91* | 80 | 93* | 80* | 88* | S | | | | 5 |
| Jhalokati | | | | | | | | 88* | 98* | 91* | 74* | 77* | S | | | | 5 |
| Patuakhali | | | | | 94* | 94* | ND | 92* | 87 | 97* | S | | | | | | 6 |
| Pirojpur | | | | | | | ND | | 79 | 93* | 92* | 58* | S | | | | 5 |
| **Khulna Division** | | | | | | | | | | | | | | | | | |
| Chuadanga | | | | | | | ND | 91* | 85 | 96* | 78* | S | | | | | 5 |
| Kushtia | | | | | | | ND | 91* | 86* | 95* | 96* | S | | | | | 5 |
| Meherpur | | | | | 71 | 73 | ND | 94* | 88 | 91* | S | | | | | | 7 |
| **Rajashahi Division** | | | | | | | | | | | | | | | | | |
| Chapainawabganj | | | | ND | 91* | 91* | ND | 91* | 83 | 92* | 89* | 94* | S | | | | 9 |
| Pabna | | | | | | | ND | 92* | 99* | 92* | 88* | S | | | | | 5 |
| Rajashahi | | | | ND | 89* | 89* | ND | 80 | 82* | 92* | S | | | | | | 7 |
| Sirajganj | | | | | | 94* | ND | 92* | 80 | 93* | 79* | S | | | | | 6 |
| **Rangpur Division** | | | | | | | | | | | | | | | | | |
| Dinajpur | | | | 93* | 92* | 93* | ND | 80 | 97* | 88* | S | | | | | | 7 |
| Kurigram | | | | 76 | 92* | 91* | 72 | 70 | 87 | 92* | 85* | 79 | 87 | S | | | 10 |
| Lalmonirhat | | 86 | 81 | 61 | 83 | 93 | 82 | 87 | 89 | 94* | 94 | 94 | 91 | S | | | 12 |
| Nilphamari | | 78 | 93* | 67 | 80 | 72 | 82 | 92* | 99* | 93* | 86 | 94 | 90 | S | | | 12 |
| Panchagarh | 93 | 83 | 82 | 75 | 95* | 94* | 84 | 86 | 92 | 95* | 96 | 97 | S | | | | 12 |
| Rangpur | | | | | 92* | 93 | 81 | 94* | 96* | 90* | 83 | 76 | 74 | 73 | ND | S | 11 |
| Thakurgaon | | 86 | 77 | 68 | 83 | 91 | 81 | 90* | 87 | 91* | 90 | 75 | S | | | | 11 |

Source: Data are provided by the LFEP. These data are also reported in Shamsuzzaman et al., 2017

ND–No data

S–Stopped MDA

* Reported coverage rates (i.e., not verified via surveys)

DHS are population-based surveys that collect data for population, health, and nutrition indicators. In Bangladesh, the surveys are conducted under the authority of the National Institute for Population Research and Training (NIPORT) of the Ministry of Health and Family Welfare (MOH&FW).

The DHS design employs a two-stage sample design. In the first stage, primary sampling units (PSU) are randomly selected from a master list maintained by the Bangladesh Bureau of Statistics. A PSU is an enumeration area with an average of 120 households. Approximately a third of the PSUs are from urban areas. In the second stage, an average of 30 households are randomly selected from each PSU. The surveys are representative of the population at the national and divisional levels and of urban and rural populations of each division. Details of sample designs can be found in the final reports of each DHS [25–29].

Because the surveys are not representative of the populations at the district level, they do not include an indicator to identify the district of each PSU. For this study, the district was identified by plotting the global positioning system coordinates of the PSUs on a 2015 map of district boundaries. This process was carried out for all five DHS using Quantum GIS, version 1.8.0-Lisboa [30,31]. The district boundaries have not changed since 1984 [32]. The unit of analysis was the individual, a woman, aged 10 to 49 years, whose last-born child was five years old or less at the time of each survey.

Data used to measure the incidence of CHE come from the HIES years 2000, 2005, and 2010. The HIES is implemented every five years by the Bangladesh Bureau of Statistics with support from the WB. As the survey title implies, it collects data on household income, expenditure, and consumption. HIES also collect data on annual medical expenses at the household level, including data on doctors' fees, hospitalization and out-patient services, medicines, maternity expenses, and health-related travel. In addition, HIES provide socio-demographic data, such as housing conditions, education, employment, health, sanitation, water supply, and electricity usage.

The HIES design also utilizes a two-stage stratified random sampling technique. Using the 2010 HIES as an example, 392 rural and 220 urban PSUs across 16 strata (six urban, six rural, and four small metropolitan areas) were randomly selected during the first stage. Twenty households were then randomly selected from each PSU to participate in the survey [33]. The HIES included 7,440 households in 2000, 10,080 households in 2005, and 12,240 households in 2010.

## Health service coverage rates

Using nationally representative DHS and Multiple Indicator Cluster Surveys, Countdown Equity Analysis Group (2008) produced a coverage gap index of eight essential health services. This index came to be known as the CCI [34–36].

The CCI includes the following indicators:

- One indicator for family planning (FP): need for FP satisfied; or contraceptive prevalence rate,

- Two indicators for maternal and newborn care: delivery assisted by a skilled birth attendant (SBA) and antenatal care received (ANC),

- Three indicators for immunization: Measles (MSL), Diphtheria, Pertussis, Tetanus (DPT–three doses), and bacille Calmette-Guerin (BCG) vaccinations,

- Two indicators for treatment of sick children: sought care for diarrhea (e.g., child given oral rehydration therapy (ORT)) and/or acute respiratory infection (ARI).

S1 File provides the definitions of each indicator as defined by the DHS Program. Equal weight is given to each indicator except for DPT because it requires three doses.

For this study, a CCI was calculated for each mother whose youngest child was born within the past five years. Data are from DHSs carried out in Bangladesh in years 1999–00, 2004, 2007, 2011, and 2014. For women, whose children did not become ill with diarrhea and an ARI within the previous two weeks of the DHS interview, the CCI was calculated with a denominator of three instead of four.

## CHE incidence

A widely accepted indicator used to measure the effect of household OOPs is CHE incidence. The total annual household OOPs is the numerator, and three different denominators can be used to measure CHE incidence: total income, total household expenditure (THE), and total household nonfood expenditure (TNFE). Of the three denominators, O'Donnell et al. argue that TNFE better distinguishes between the wealthier and the less well-off because most food purchases are nondiscretionary and therefore, less sensitive to unplanned health expenses [37].

The economic burden borne by those of the upper wealth quintiles is proportionately smaller because their capacity to pay is higher. Therefore, measuring CHE incidence has three important limitations: it ignores the additional economic hardships due to lost earnings; it ignores those households that cannot afford the health expense, forego treatment, and

therefore, suffer greater loss than those incurring CHE; and it counts all households equally, irrespective of whether the payments are made by the wealthy or less well-off [37].

This study measured CHE incidence using annual THE and annual TNFE as the denominators. The recall period for all inpatient and outpatient medical expenses was the 12 months prior to the survey interview. OOPs of the last 12 months for all individuals in the household were summed up and divided by the denominators. A range of thresholds was used in the analysis: 5%, 10%, and 15% thresholds when THE is the denominator; and 20%, 25%, 30%, and 40% thresholds when TNFE is the denominator. These thresholds are consistent with published literature [37,38]. S2 File provides the methodology for calculating OOP, THE, and TNFE [37,39]. The unit of analysis for this outcome variable is the household.

## Control variables

One of the measures of a well-functioning health system is equity across different groups [18]. Therefore, the analysis includes the effects of living in rural areas, belonging to lower wealth quintiles, and having less than a primary level education. These are the determinants used in the literature [35,38,40]. Other control variables include employment and marital status. Given the different levels of analysis (household vs. individual) and data available in the DHS and HIES datasets, the control variables are different for each outcome variable. Also, categorical variables were recoded to create binary variables (Table 3).

In this study, wealth quintiles, based on an index of household assets, are used instead of income as a measure of standard of living because, in low-income countries like Bangladesh, income is difficult to measure. For example, subsistence farmers may receive income intermittently and/or may receive in-kind products or services for their crop. These quintiles are based on a wealth index calculated using principal components analysis (PCA).

DHS collects data on several assets such as quality of housing, sanitation, ownership of durable, and household goods to calculate the wealth index. The wealth index and wealth

**Table 3. Control variables and reference categories for each outcome variable.**

| Categorical/Count Variables | Binary Variables | Reference Category | Outcome Variable | |
|---|---|---|---|---|
| | | | Health service coverage rate | Incidence of CHE |
| Age of Mother | 10–14 years, 15–19 years, 20–24 years, 25–29 years, 30–34 years, 35–39 years, and 40–44 years | 45–49 years | ✓ | |
| Employment status* | Unemployed | Employed | ✓ | |
| Employment status† | Employed | Unemployed | | ✓ |
| Gender | Female | Male | | ✓ |
| Highest level of education attained by head of household | Some primary, primary, some secondary, secondary, and higher education | No education | | ✓ |
| Highest level of education attained by mother | No education, primary, and secondary | Higher education | ✓ | |
| Marital status of mother or head of household | Married | Not married (e.g., never married, widowed, divorced, or separated) | ✓ | ✓ |
| Number of chronically ill household members | One member; two members; three or more members | Zero members | | ✓ |
| Number of household members | Not applicable (count variable) | | | ✓ |
| Residence | Rural | Urban | ✓ | ✓ |
| Wealth quintile | 1st quintile, 2nd quintile, 4th quintile, 5th quintile | 3rd quintile | ✓ | ✓ |

* The DHS defined employed as "whether the respondent is currently working."

†The HIES defined employed as "working for a livelihood in the seven days prior to the survey."

quintile for each survey respondent were included in the DHS survey data sets. The PCA function of Stata was used to create an asset index at the household level for each of the 16 strata of the HIES. The separate asset indices account for the economic and cultural differences and the urban and rural environments of each stratum. For example, livestock may contribute more to an asset index of a rural household than an urban household, and the separate indices produced using PCA account for this difference. The asset index for this study includes housing materials and features, access to water and sanitation facilities, livestock, and durable goods (e.g., radio, refrigerator, and television)[37,41]. S2 File describes how CHE incidences were calculated and provides a list of assets included in the index.

## Study design

A DiD model was used to measure the effects of the LFEP on essential health service coverage rates and CHE incidence. The study design accounts for other time-dependent trends by using a control group (i.e., hypo- and non-LF-endemic districts) that is experiencing the same trends but not the intervention [42,43].

## Estimating equations

Three DiD equations were estimated for each outcome variable: a parsimonious specification; a specification including control variables; and a specification including control variables and district-level fixed-effects to control for the time-invariant characteristics of each district that may bias the outcome (e.g., cultural norms, agro-ecological characteristics). The estimating equations for health service coverage rates are presented below. Estimating equations for CHE incidence and definitions of the terms are presented in S3 File.

$$Y_{it} = \beta_0 + \beta_1 T_i + \beta_2 DiD\ Estimator_{it} + \beta_3 Ended\ Treatment_{it} + \alpha Year\ Fixed\ Effects_t + \varepsilon_{it} \tag{Eq 1}$$

$$Y_{it} = \beta_0 + \beta_1 T_i + \beta_2 DiD\ Estimator_{it} + \beta_3 Ended\ Treatment_{it} + \alpha Year\ Fixed\ Effects_t + \beta_4 5^{th}\ quintile_{it} + \beta_5 4^{th}\ quintile_{it} + \beta_6 2^{nd}\ quintile_{it} + \beta_7 1^{st}\ quintile_{it} + \beta_8 Rural_{it} + \beta_9 No\ Education_{it} + \beta_{10} Primary_{it} + \beta_{11} Secondary_{it} + \beta_{12} 15-19\ years_{it} + \beta_{13} 20-24\ years_{it} + \beta_{14} 25-29\ years_{it} + \beta_{15} 30-34\ years_{it} + \beta_{16} 35-39\ years_{it} + \beta_{17} 40-44\ years_{it} + \beta_{18} Married_{it} + \beta_{19} Unemployed_{it} + \varepsilon_{it} \tag{Eq 2}$$

$$Y_{ijt} = \beta_0 + \beta_1 T_i + \beta_2 DiD\ Estimator_{it} + \beta_3\ Ended\ Treatment_{it} + \alpha Year\ Fixed\ Effects_t + \beta_4 5^{th}\ quintile_{it} + \ldots + \beta_{19} Unemployed_{it} + \gamma District\ Fixed\ Effects_i + \varepsilon_{ijt} \tag{Eq 3}$$

To account for the different MDA start and end dates of each treatment district (see Table 1), the DiD Estimators in Eqs 1–3 are coded as the proportion of years that the MDAs were implemented from one survey year to the next. For example, if MDAs were implemented in a treatment district for only one of the four years between surveys, the DiD estimator was 0.25. Otherwise, the DiD estimator was zero (0). *Year Fixed Effects*$_t$, capture secular changes in health service coverage rates across Bangladesh. An "ended treatment" term is included to differentiate control districts from treatment districts no longer implementing MDAs. To allow for serial correlation in outcomes within a district all standard errors are clustered by district.

Marginal effects (MFX) were calculated to investigate whether disadvantaged populations are accessing essential health services or experiencing CHEs at the same rate as non-disadvantaged populations. Disadvantaged populations are individuals living in rural areas, with less formal education, and of lower wealth quintiles. [35,38,40].

We also tested for heterogenous treatment effects between urban and rural populations and between the upper two wealth quintiles and lower two wealth quintiles as the program may have served to close health disparities along these socio-economic axes. We adapted our three main specifications above by adding two terms: a control for being in the disadvantaged group of interest (e.g., rural), and an interaction term between the DiD estimator and an indicator for being in the disadvantaged group of interest. The coefficient of interest is on the latter term and gives the differential impact of the program in the disadvantaged group. Eqs 6 and 9 are the full models with the rural and wealth quintile interaction terms added, respectively. Eqs 4, 5, 7, and 8 for health service coverage rates and CHE are presented in S3 File.

$$Y_{ijt} = \beta_0 + \beta_1 T_i + \beta_2\, DiD\ Estimator_{it} + \beta_3\, (DiD\ Estimator * Rural)_{it} + \beta_4$$
$$Rural_{it} + \beta_5 Ended\ Treatment_{it} + \alpha Year\ Fixed\ Effects_{it} + \ldots + \beta_{20} Unemployed_{it} \qquad \text{Eq 6}$$
$$+ \gamma District\ Fixed\ Effects_i + \varepsilon_{ijt}$$

$$Y_{ijt} = \beta_0 + \beta_1 T_i + \beta_2 DiD\ Estimator_{it} + \beta_3 (DiD\ Estimator * Low\ wealth$$
$$quintiles)_{it} + \beta_4 Low\ wealth\ quintiles_{it} + \beta_5 Ended\ Treatment_{it} + \alpha Year\ Fixed \qquad \text{Eq 9}$$
$$Effects_{it} + \ldots + \beta_{17} Unemployed_{it} + \gamma District\ Fixed\ Effects_i + \varepsilon_{ijt}$$

We also tested a specification limited to a control group that may more closely resemble the treated districts. For this check, we included only the 15 hypo-endemic districts along with the treated districts. All analyses were conducted using Stata version 15.1.

## Results

### Health-seeking behaviors

For the individuals that reported being ill in the 30 days prior to the HIES interview, the most common complaint for all survey years was fever ($> 55\%$). The second most common complaint was pain ($>10\%$). Individuals in treatment districts were less likely to report being ill in the baseline year but more likely to in the subsequent survey years. Individuals in treatment districts were also more likely to report suffering from a chronic illness (p-value $< 0.00$). The most common chronic illnesses across all groups and years were "gastric/ulcer" ($> 25\%$), "other" ($> 15\%$), "arthritis/rheumatism" ($> 12\%$). The HIES did not ask respondents if they sought medical treatment for their chronic conditions.

Of those who reported being ill, more than 70% sought medical care. Individuals in control districts were more likely to seek medical care (p-value $< 0.00$). The most common reasons for not seeking medical care were that the illness was not considered serious or that it cost too much. Only the 2010 HIES included concerns about quality of care as a reason for not seeking treatment. Less than 1% of the survey population selected this reason for not seeking medical treatment.

### Effects of the LFEP on health service coverage rates

For both the treatment and control districts, Table 4 presents socio-demographic characteristics, the CCI scores, and the coverage rates of each health service included in the CCI of women whose last-born child was five years or less for the baseline DHS year 1999–2000.

**Table 4. Demographic characteristics and CCI of women with children 5 years or less by treatment and control districts for Bangladesh DHS, baseline year 1999–2000.**

| Socio-demographic characteristics and tracer indicators of CCI | Treatment n (%) | Control n (%) | p- value |
|---|---|---|---|
| Population surveyed—women with children < = 5 years (N) | 1,485 (26.90) | 4,035 (73.10) | |
| Residence | | | |
| Urban | 363 (24.44) | 1,130 (28.00) | 0.008 |
| Rural | 1,122 (75.56) | 2,911 (72.14) | |
| Wealth index | | | |
| First (lowest 20%) | 323 (21.75) | 883 (21.88) | 0.000 |
| Second | 367 (24.71) | 747 (18.51) | |
| Third | 313 (21.08) | 733 (18.17) | |
| Fourth | 231 (15.56) | 724 (17.94) | |
| Fifth (highest 20%) | 251 (16.90) | 948 (23.49) | |
| Education | | | |
| No education | 609 (41.01) | 1,773 (43.94) | 0.000 |
| Primary | 495 (33.33) | 1,104 (27.36) | |
| Secondary | 321 (21.61) | 935 (23.17) | |
| Higher education | 60 (4.04) | 223 (5.53) | |
| Age (years) | | | |
| 10–14 | 9 (0.61) | 7 (0.17) | 0.000 |
| 15–19 | 262 (17.64) | 587 (14.32) | |
| 20–24 | 462 (31.11) | 1,112 (27.56) | |
| 25–29 | 381 (25.66) | 1,118 (27.71) | |
| 30–34 | 207 (13.94) | 714 (17.70) | |
| 35–39 | 111 (7.47) | 321 (8.05) | |
| 40–44 | 42 (2.83) | 140 (3.47) | |
| 45–49 | 11 (0.74) | 41 (1.02) | |
| Marital status | | | |
| Married | 1,451 (97.71) | 3,915 (97.03) | 0.016 |
| Widowed | 21 (1.41) | 41 (1.02) | |
| Divorced | 4 (0.27) | 12 (0.30) | |
| Separated | 16 (0.29) | 67 (1.66) | |
| Employment status | | | |
| Employed | 295 (19.87) | 707 (17.52) | 0.045 |
| Unemployed | 1,190 (80.13) | 3,328 (82.48) | |
| Composite coverage index | | | |
| Family planning needs met | 1,144 (80.73) | 2,987 (77.71) | 0.018 |
| Antenatal care by skilled provider | 534 (40.06) | 1,334 (36.03) | 0.009 |
| Delivery assisted by a skilled health professional | 325 (24.36) | 1,041 (28.12) | 0.008 |
| BCG immunization received | 1,199 (90.35) | 3,247 (88.14) | 0.029 |
| 3 doses of DPT immunization received | 921 (69.40) | 2,532 (68.71) | 0.640 |
| Measles immunization received | 901 (67.90) | 2,500 (67.84) | 0.971 |
| Sought care for ARI (N = children with cough, fever, and rapid breathing in last 2 weeks) | 229 (61.23) | 776 (65.54) | 0.129 |
| Sought treatment for diarrhea (N = number of children with diarrhea in last 2 weeks) | 35 (46.05) | 172 (55.66) | 0.132 |
| CCI–population | 60.21 | 60.93 | 0.001 |
| CCI–mean of individuals | 57.36 | 57.01 | |

The Chi-Square Test p-values of all socio-demographic variables are significant (p-value < 0.05), which means the two groups are statistically different from one another at the time of the baseline survey. The survey population living in treatment districts are more likely

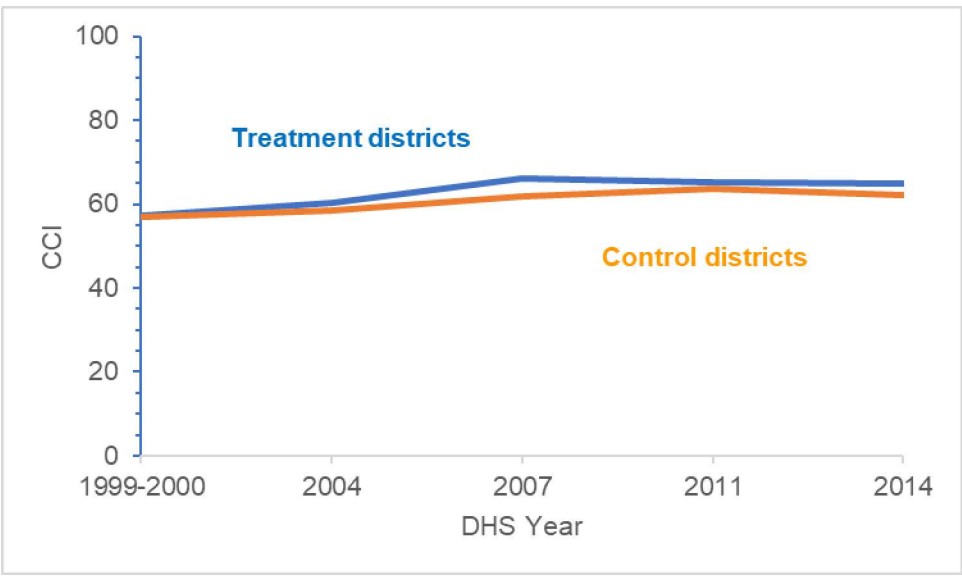

**Fig 2. CCI trends for treatment and control districts from 2000–14.**

to live in rural areas and belong to lower wealth quintiles. However, they are also more likely to have completed at least primary education and to be employed. The results justify controlling for these variables in the estimating equations. Regarding outcome variables, the survey population living in treatment districts were more likely to have their family planning needs met, receive ANC, and have their children vaccinated against BCG. The coverage rates for the other health services between the two groups were not statistically different (p-value > 0.05).

In 2000, the mean CCI score of the population living in treatment districts was 0.35 percentage point above the population living in control districts. This difference was significant (p-value < .05). By 2014, the mean CCI score was 2.73 percentage points higher in treatment districts than in control districts. Increases in coverage rates for all health services that make up the CCI were higher in treatment districts except for BCG immunization. Fig 2 compares the trends in CCI scores between the treatment and control districts.

However, the mean DiD between the control and treatment districts was not statistically significant from zero (p-value > 0.05). For Eq 3, individuals living in treatment districts had a CCI score that was 0.146 percentage point higher than individuals living in control districts. Regarding disadvantaged populations, when compared to the reference category (Table 3), individuals who were older or of lower wealth quintiles had less formal education, or were living in rural areas had lower coverage rates. The results of the DiD analyses for the CCI score and each essential health service are provided in S4 File.

Table 5 presents the essential health service coverage rate DiD estimator values for Eqs 3, 6, and 9 for both the original (i.e., 15 hypo- and 30 non-endemic) and smaller (15 hypo-endemic) control groups of districts. Eq 3 is the original full model with all control variables plus the district fixed effects term. It is included in Table 5 to facilitate comparison. Eqs 6 and 9 are also the full models and test for heterogeneous treatment effects between the rural and urban populations and between populations of lower and higher wealth quintiles, respectively, in treatment districts. The results for both control groups are similar (i.e., results generally do not switch from significant to non-significant or negative to positive and vice versa across models). The models are robust to the different specification of a smaller control group. Rural and lower wealth quintile populations are less likely than urban and wealthy populations,

**Table 5. Summary of DiD analyses of health service coverage rates, including heterogeneous treatment effects using both control groups.**

| | Control – 15 hypo-endemic plus 30 non-endemic districts | | | Control – 15 hypo-endemic districts | | |
|---|---|---|---|---|---|---|
| | Eq 3 | Eq 6 Rural | Eq 9 LWQ | Eq 3 | Eq 6 Rural | Eq 9 LWQ |
| **CCI** | | | | | | |
| DiD Estimator | 0.00146 (0.00884) | -0.00120 (0.0114) | -0.000207 (0.00912) | 0.00367 (0.0107) | 0.0131 (0.0124) | 0.00289 (0.0109) |
| DiD Estimator X Rural or DID Estimator X LWQ | | 0.00379 (0.0105) | 0.00369 (0.0126) | | -0.0136 (0.0119) | -0.000329 (0.0127) |
| **FP needs satisfied** | | | | | | |
| DiD Estimator | 0.0104 (0.0164) | -0.0105 (0.0197) | 0.00522 (0.0179) | 0.0107 (0.0170) | 0.00790 (0.0185) | 0.00748 (0.0182) |
| DiD Estimator X Rural or DID Estimator X LWQ | | 0.0284+ (0.0156) | 0.0196 (0.0172) | | 0.00392 (0.0149) | 0.0120 (0.0164) |
| **Antenatal care by skilled provider** | | | | | | |
| DiD Estimator | 0.00468 (0.0179) | 0.0207 (0.0278) | 0.00267 (0.0183) | 0.00651 (0.0222) | 0.0442 (0.0298) | 0.00689 (0.0233) |
| DiD Estimator X Rural or DID Estimator X LWQ | | -0.0213 (0.0288) | -0.00267 (0.0242) | | -0.0505+ (0.0284) | -0.00994 (0.0249) |
| **Delivery assisted by an SBA** | | | | | | |
| DiD Estimator | 0.0250+ (0.0137) | 0.0747*** (0.0217) | 0.0353* (0.0152) | 0.00692 (0.0184) | 0.0615* (0.0291) | 0.0168 (0.0201) |
| DiD Estimator X Rural or DID Estimator X LWQ | | -0.0745** (0.0262) | -0.0539** (0.0193) | | -0.0828** (0.0282) | -0.0524** (0.0198) |
| **BCG immunization** | | | | | | |
| DiD Estimator | 0.00586 (0.00991) | 0.00253 (0.0110) | 0.00478 (0.00950) | 0.00740 (0.0104) | 0.00611 (0.0118) | 0.00750 (0.00981) |
| DiD Estimator X Rural or DID Estimator X LWQ | | 0.00450 (0.0128) | 0.00243 (0.0129) | | 0.00177 (0.0135) | -0.000563 (0.0118) |
| **3 doses of DPT immunization** | | | | | | |
| DiD Estimator | 0.00484 (0.0176) | -0.00568 (0.0180) | 0.00459 (0.0182) | 0.0106 (0.0223) | 0.00767 (0.0226) | 0.0124 (0.0224) |
| DiD Estimator X Rural or DID Estimator X LWQ | | 0.0146 (0.0193) | -0.00155 (0.0163) | | 0.00410 (0.0208) | -0.00792 (0.0162) |
| **Measles immunization** | | | | | | |
| DiD Estimator | 0.00675 (0.0166) | 0.0161 (0.0249) | -0.00434 (0.0181) | 0.0130 (0.0207) | 0.0268 (0.0289) | 0.00314 (0.0215) |
| DiD Estimator X Rural or DID Estimator X LWQ | | -0.0130 (0.0251) | 0.0425* (0.0195) | | -0.0194 (0.0264) | 0.0398* (0.0200) |
| **Sought care for ARI** | | | | | | |
| DiD Estimator | -0.00688 (0.0267) | -0.000175 (0.0461) | -0.00452 (0.0273) | -0.0181 (0.0344) | -0.00208 (0.0524) | -0.0184 (0.0368) |
| DiD Estimator X Rural or DID Estimator X LWQ | | -0.00943 (0.0411) | -0.0236 (0.0319) | | -0.0224 (0.0412) | -0.0230 (0.0325) |
| **Sought treatment for diarrhea** | | | | | | |
| DiD Estimator | 0.0350 (0.0511) | -0.0410 (0.0665) | 0.0383 (0.0560) | 0.0504 (0.0680) | 0.00157 (0.0782) | 0.0567 (0.0712) |
| DiD Estimator X Rural or DID Estimator X LWQ | | 0.120 (0.0727) | -0.0227 (0.0580) | | 0.0783 (0.0771) | -0.0336 (0.0557) |

+ p < .1

* p < .05

** p<0.01

*** p<0.001 (Clustered standard errors in parentheses), LWQ – Lowest wealth quintiles

respectively, to benefit from the treatment effect on delivery assisted by an SBA. Lower wealth quintile populations were more likely than higher wealth quintile populations to have their children immunized against measles.

## Effects of the LFEP on CHE Incidence

Table 6 summarizes the socio-demographic characteristics of households in treatment and control districts for baseline HIES year 2000. The Chi-Square Test p-values of all socio-demographic variables, except marital status, are significant (p-value < 0.05), which means the two groups are statistically different from one another at the time of the baseline survey. Households in the treatment districts were less likely to live in urban areas or to attain more than a primary-level education. They were also more likely to have at least one chronically ill member. Regarding outcome variables, households living in treatment districts were also more likely to experience a CHE at all thresholds.

When THE is used as the denominator, households in both treatment and control districts experienced downward trends in incidence of CHE between years 2000 and 2010. However, when TNFE is used as the denominator, households in treatment districts experienced a downward trend in incidence of CHE while households in control districts experienced a slight uptick in incidence. Figs 3 and 4 illustrate these trends for the THE and TNFE denominators, respectively.

The DiD estimator was significant (the mean DiD between the control and treatment districts was statistically significant from zero (p-value < 0.05)) in Eqs 1 and 2, and 3 for all thresholds except for 5% THE and 20% TNFE; the lowest threshold for each denominator. For examples, when using 10% of THE as a threshold, households of treatment districts were more than 9.6 percentage points less likely to experience a CHE, and when using 25% of TNFE as a threshold, the results were similar: households in treatment districts were more than 6.9 percentage points less likely to experience a CHE (see Eq 3 results in Table 7). S5 File presents the results of DiD analyses for all three estimating equations for each CHE threshold.

Lower wealth quintile households were neither more likely nor less likely to experience a CHE except at the threshold of 20% TNFE. Households in rural areas were less likely to experience a CHE at the higher thresholds of 15% THE and 40% TNFE. Households with chronically ill members were more likely to experience a CHE at the lower thresholds but less likely at the higher thresholds.

Table 7 presents the CHE incidence DiD estimator values for Eqs 3 and 6, and Eq 9. For Eq 3, the DiD estimators are similar in significance for both control groups. For Eq 6 and Eq 9, the results were mixed across the thresholds. Rural households of treatment districts, when compared to both groups of control districts, were not less likely to experience a CHE. However, none of the DiD estimator values were significant except at the 20% TNFE threshold of the smaller control group. At the 5% THE threshold level, lower wealth quintile households in the smaller control group were 4.34 percentage points less likely to experience a CHE. At the larger thresholds, the DiD estimators for lower wealth quintile households when compared to both control groups were insignificant.

## Discussion

The results of this study add to the evidence of the effects of DSPs on health systems. It is one of the few that focuses on an NTD program whereas most of the other studies on this topic focus on HIV/AIDS, tuberculosis, or malaria programs. In addition, whereas previous studies used cross-sectional surveys or observations in targeted districts to assess the effects of DSPs [6,13,44], this study used data from several cross-sectional population-based surveys to compare outcome tracer indicators of treatment and control districts.

**Table 6.  Sociodemographic characteristics of and CHE incidence in households in treatment and control districts for Bangladesh HIES baseline year, 2000.**

| Household Characteristics | Treatment n (%) | Control n (%) | p-value |
|---|---|---|---|
| Households (N) | 2,021 (27.16) | 5,419 (72.84) | |
| Residence | | | |
| Urban | 598 (29.59) | 1,799 (32.83) | 0.008 |
| Rural | 1,432 (70.41) | 3,640 (67.17) | |
| Asset Index | | | |
| First (Lowest 20%) | 452 (22.38) | 1041 (19.21) | 0.007 |
| Second | 420 (20.79) | 1,069 (19.73) | |
| Third | 394 (19.50) | 1,093 (20.17) | |
| Fourth | 389 (19.26) | 1,099 (20.28) | |
| Fifth (Highest 20%) | 365 (18.07) | 1,117 (20.61) | |
| HH Education level | | | |
| No education | 1,157 (57.25) | 3,046 (56.21) | 0.000 |
| Primary | 320 (15.83) | 770 (14.21) | |
| Secondary | 412 (20.39) | 1,079 (19.19) | |
| Higher | 132 (6.53) | 524 (9.67) | |
| HH Gender | | | |
| Female | 142 (7.03) | 524 (9.67) | 0.000 |
| Male | 1,879 (92.97) | 4,895 (90.33) | |
| HH marital status | | | |
| Married | 1,824 (90.25) | 4,865 (89.78) | 0.295 |
| Never married | 57 (2.82) | 151 (2.79) | |
| Widowed | 129 (6.38 | 340 (6.27) | |
| Divorced | 3 (0.15) | 16 (0.30) | |
| Separated | 8 (0.40) | 47 (0.87) | |
| Number of chronically ill HH members | | | |
| 0 | 907 (44.88) | 2,712 (50.05) | 0.000 |
| 1 | 657 (32.51) | 1,613 (29.77) | |
| 2 | 332 (16.43) | 776 (14.32) | |
| >2 | 125 (6.18) | 317 (5.85) | |
| Median HH income and expenditures (Bangladesh Taka) | | | |
| Income | 36,000.00 | 47,908.00 | |
| Total expenditure | 16,834.57 | 29,008.36 | |
| Nonfood expenditure | 11,852.00 | 17,224.00 | |
| OOP medical expenses | 530.00 | 800.00 | |
| CHE–OOP/THE | | | |
| 5% | 913 (45.18) | 2,047 (37.77) | 0.000 |
| 10% | 554 (27.41) | 1,132 (20.89) | 0.000 |
| 15% | 410 (20.29) | 815 (15.04) | 0.000 |
| CHE–OOP/TNFE | | | |
| 20% | 393 (19.45) | 802 (14.80) | 0.000 |
| 25% | 364 (18.01) | 697 (12.86) | 0.000 |
| 30% | 331 (16.38) | 632 (11.66) | 0.000 |
| 40% | 302 (14.94) | 549 (10.13) | 0.000 |

The CCI and incidence of CHE are well-established indicators for measuring the strength of health systems and ultimately, progress toward UHC at the national level. This study showed how the CCI and CHE incidence, calculated using data from DHS and HIES,

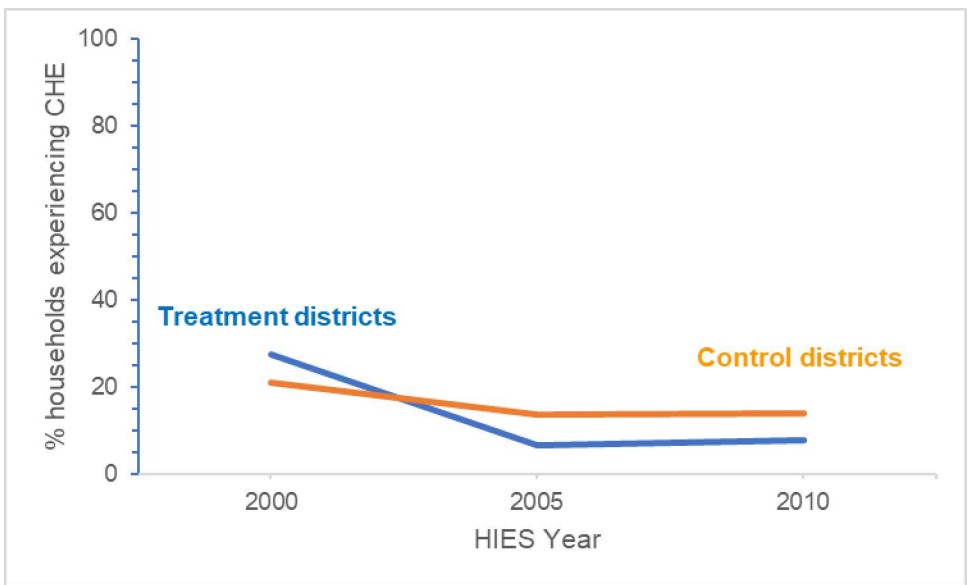

**Fig 3. Trends in incidence of households in treatment and control districts experiencing CHEs, using 10% of THE as a threshold.**

respectively, can be used to quantify DSPs' effects on health systems and equity in reaching disadvantaged populations at the district level. Similar surveys are implemented in other low- and middle-income countries, and the data from these surveys are typically available to the public. Therefore, this study can be easily replicated in other countries with minimal cost. Alternative sources are data collected via District Health Information System-2, which are collected at more frequent intervals. These data can be used to monitor the effect of a DSP soon after implementation by measuring changes in service coverage.

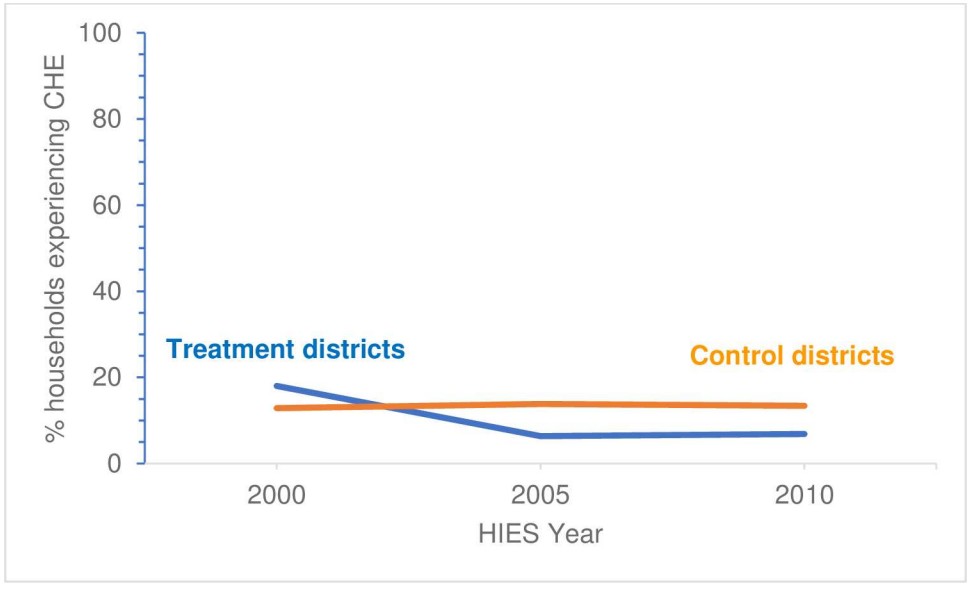

**Fig 4. Trends in incidence of households in treatment and control districts experiencing CHEs, using 25% of TNFE as a threshold.**

**Table 7. Summary of DiD analyses of incidence of CHE at all thresholds, including heterogeneous treatment effects using both control groups.**

| | Control – 15 hypo-endemic plus 30 non-endemic districts | | | Control – 15 hypo-endemic districts | | |
|---|---|---|---|---|---|---|
| | Eq 3 | Eq 6 Rural | Eq 9 LWQ | Eq 3 | Eq 6 Rural | Eq 9 LWQ |
| **5% THE** | | | | | | |
| DiD Estimator | -0.0687 (0.0446) | -0.0796+ (0.0411) | -0.0616 (0.0436) | -0.0921+ (0.0498) | -0.115** (0.0431) | -0.0837+ (0.0490) |
| DiD Estimator X Rural or DID Estimator X LWQ | | 0.0173 (0.0253) | -0.0385+ (0.0206) | | 0.0359 (0.0280) | -0.0434* (0.0202) |
| **10% THE** | | | | | | |
| DiD Estimator | -0.0966** (0.0343) | -0.102** (0.0364) | -0.0927** (0.0343) | -0.118** (0.0433) | -0.131** (0.0431) | -0.113** (0.0433) |
| DiD Estimator X Rural or DID Estimator X LWQ | | 0.00860 (0.0293) | -0.0207 (0.0171) | | 0.0224 (0.0311) | -0.0262 (0.0172) |
| **15% THE** | | | | | | |
| DiD Estimator | -0.0964** (0.0326) | -0.109** (0.0391) | -0.0958** (0.0329) | -0.114** (0.0417) | -0.133** (0.0451) | -0.112** (0.0418) |
| DiD Estimator X Rural or DID Estimator X LWQ | | 0.0211 (0.0318) | -0.00377 (0.0191) | | 0.0318 (0.0335) | -0.00876 (0.0195) |
| **20% TNFE** | | | | | | |
| DiD Estimator | -0.0465 (0.0372) | -0.0748+ (0.0418) | -0.0441 (0.0368) | -0.0549 (0.0470) | -0.0947* (0.0478) | -0.0516 (0.0470) |
| DiD Estimator X Rural or DID Estimator X LWQ | | 0.0438 (0.0273) | -0.0124 (0.0171) | | 0.0623* (0.0283) | -0.0159 (0.0171) |
| **25% TNFE** | | | | | | |
| DiD Estimator | -0.0692* (0.0347) | -0.0990* (0.0406) | -0.0697* (0.0346) | -0.0783+ (0.0434) | -0.119** (0.0459) | -0.0776+ (0.0435) |
| DiD Estimator X Rural or DID Estimator X LWQ | | 0.0461 (0.0292) | 0.00150 (0.0167) | | 0.0633* (0.0306) | -0.00258 (0.0167) |
| **30% TNFE** | | | | | | |
| DiD Estimator | -0.0773* (0.0339) | -0.109* (0.0431) | -0.0815* (0.0335) | -0.0868* (0.0419) | -0.129** (0.0473) | -0.0896* (0.0417) |
| DiD Estimator X Rural or DID Estimator X LWQ | | 0.0490 (0.0317) | 0.0178 (0.0152) | | 0.0659* (0.0326) | 0.0130 (0.0154) |
| **40% TNFE** | | | | | | |
| DiD Estimator | -0.0798* (0.0351) | -0.0975* (0.0453) | -0.0840* (0.0357) | -0.0911* (0.0419) | -0.116* (0.0487) | -0.0942* (0.0425) |
| DiD Estimator X Rural or DID Estimator X LWQ | | 0.0287 (0.0319) | 0.0171 (0.0161) | | 0.0416 (0.0330 | 0.0134 (0.0167) |

+ p < .1

* p < .05

** p<0.01

*** p<0.001 (Clustered standard errors in parentheses), LWQ – Lower wealth quintiles.

Using the CCI to measure the effects of the LFEP revealed that it did not negatively or positively affect health service coverage rates. CCI scores increased for both populations. However, utilization of services did not necessarily increase for those of the lower wealth quintiles, that had less formal education, or were living in rural areas. Using CHE incidence as an outcome measure, it can be said that there is a positive association. After at least five years of LFEP interventions, households in treatment districts had a lower incidence of CHE at all thresholds except 5% THE and 20% TNFE, which were the lowest thresholds for each denominator. One

possible reason for the increase in financial protection is positive interactions with health staff during the LFEP interventions. Another possible reason is that the health staff may take the opportunity to screen individuals for healthcare needs and make referrals, which in turn, could decrease CHE incidence because individuals receive care before the condition progresses and may be less likely to seek care from untrained practitioners. These reasons are speculation, and more research is needed.

Interest in assessing the effect of OOPs has increased because financial protection from CHEs is one of the underpinnings of UHC [45]. It is not surprising that there are already a handful of recent publications that measure the incidence of CHE in Bangladesh, some using the 2010 HIES. Studies using the HIES 2010 dataset explored the incidence of CHE at the national level among households in the different wealth quintiles [23,38,46]. Results revealed that rural households and households in the lower wealth quintiles experienced a higher concentration of CHEs. Results of this present study were different because wealth quintiles were calculated using different methods, and MFX, not concentration indices, were used to compare differences between the higher and lower wealth quintiles. Also, in this study, households in the lower and upper wealth quintiles were compared to those of the middle quintile.

Results indicate that households with chronically ill members or households headed by females had lower incidence of CHE. These results may be due to one of the limitations of using CHE as an indicator to measure the effect of OOPs; methods used to calculate CHEs do not take into consideration poorer households that forego treatment [37]. This conclusion assumes that households with chronically ill members or headed by females are among the lower wealth quintiles.

Individuals in treatment districts were more likely to report being ill in the 30 days prior to the survey and more likely to seek treatment in 2010 than they were in 2000. More research is needed to determine if the increase in self-reported illness and change in treatment-seeking behaviors are the result of the LFEP.

The outcomes presented in this paper are different from those presented in Cavelli et al. [6]. It is plausible that the positive outcome reflects how well the LFEP was integrated into the health system. However, standard definitions and measurements of integration are difficult to establish, but best practices can continue to be documented [10–13].

Several limitations need to be considered.

1. Secondary data. The DHS and HIES are representative of populations living in urban and rural areas and at the national and divisional levels. These surveys are not representative of populations at the district level.

2. Parallel trends. The key assumption in the DiD study design is that treatment and control groups experience similar growth in outcomes in the absence of treatment (parallel trends). Unfortunately, we could not directly test this assumption as the districts of each PSU could not be identified prior to the 1999–2000 DHS.

3. Bundled shocks. We did not have data on other DSPs like malaria and Kala-azar control programs or the work of non-governmental or community-based organizations. If these programs started in treatment districts at the same time as LFEP then we would mistakenly have included the effects of these programs with our estimates of the impacts of LFEP. If, however, these other projects were not coincident with LFEP in treatment districts then we have estimated a lower bound of the impacts of LFEP as these programs may have had some impacts in the control group.

4. Validity of measures. This study was not designed to prove the validity of the outcome measures (i.e., whether or not CCI and CHE incidence are valid indicators to measure the

strength of a health system). These tracer indicators are established by WHO and WB and widely used by researchers to measure the strength of health systems. The outcome variable, CCI, is intended to compare large amounts of data from multiple secondary sources across countries. It is simplistic. Regarding CHE, there are several limitations to using it to measure the impact of OOPs, which is why two separate measures with a range of thresholds were used.

5. Omitted variables. The full regression model includes the control variables commonly presented in the literature. However, there are other variables that could be included in the regression equations, such as other barriers to accessing healthcare (e.g., distance to clinic), but they were not measured in the surveys. While our specification with district fixed effects may capture their impacts if they do not change, our specification may be biased if there are substantive changes in these factors over the study period.

6. Spillover effects. Effects of the LFEP may spillover to neighboring control districts. This will bias our estimates of LFEP toward zero and make us less likely to detect an impact.

7. External validity. The purpose of the study is to assess the effects of LFEP on Bangladesh's health system. The results cannot be inferred to the effects of LFEPs and DSPs on health systems of other countries. There are, however, several other countries where the methodology presented in this study can be applied because LFEPs present natural experiments.

## Supporting information

**S1 File. Calculating the Composite Coverage Index.**
(DOCX)

**S2 File. Calculating incidence of catastrophic health expenditures.**
(DOCX)

**S3 File. Estimating equations.**
(DOCX)

**S4 File. Results of Essential Health Service Coverage Rates DiD Analyses.**
(DOCX)

**S5 File. Results of catastrophic health expenditure incidence DiD Analyses.**
(DOCX)

## Acknowledgments

Dr. Mujib Rahman and Dr. ASM Sultan Mahmood of the Lymphatic Filariasis Elimination Program, Bangladesh Ministry of Health and Family Welfare, for providing treatment data, photographs, and insight into the implementation of the LFEP.

Mr. Md. Karamat Ali of the Bangladesh Bureau of Statistics for his guidance and support in accessing HIES data.

## Author Contributions

**Conceptualization:** Kimberly M. Koporc.

**Data curation:** Kimberly M. Koporc.

**Formal analysis:** Kimberly M. Koporc.

**Investigation:** Kimberly M. Koporc.

**Methodology:** Kimberly M. Koporc, David R. Hotchkiss, Charles F. Stoecker, Deborah A. McFarland, Thomas Carton.

**Supervision:** David R. Hotchkiss, Charles F. Stoecker, Deborah A. McFarland, Thomas Carton.

**Validation:** Kimberly M. Koporc.

**Writing – original draft:** Kimberly M. Koporc.

**Writing – review & editing:** David R. Hotchkiss, Charles F. Stoecker, Deborah A. McFarland, Thomas Carton.

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
