## [Decision Letter · Decision Letter 0]

7 Sep 2021

Dear Dr. Koporc,

Thank you very much for submitting your manuscript "Assessing the effects of disease-specific programs on health systems: An analysis of the Bangladesh Lymphatic Filariasis Elimination Program’s impacts on health service coverage and catastrophic health expenditure" for consideration at PLOS Neglected Tropical Diseases. As with all papers reviewed by the journal, your manuscript was reviewed by members of the editorial board and by several independent reviewers. The reviewers appreciated the attention to an important topic. Based on the reviews, we are likely to accept this manuscript for publication, providing that you modify the manuscript according to the review recommendations. 

Sincerely,

Alberto Novaes Ramos Jr

Associate Editor

Francesca Tamarozzi

Deputy Editor

Reviewer's Responses to Questions

**Key Review Criteria Required for Acceptance?**

**Methods**

-Are the objectives of the study clearly articulated with a clear testable hypothesis stated?

-Is the study design appropriate to address the stated objectives?

-Is the population clearly described and appropriate for the hypothesis being tested?

-Is the sample size sufficient to ensure adequate power to address the hypothesis being tested?

-Were correct statistical analysis used to support conclusions?

-Are there concerns about ethical or regulatory requirements being met?

Reviewer #1: The methods are appropriate for the research question - the Difference In Difference (DiD) indicator used is a good way to control for internal confounding within health districts and comparison to the control group(s) helps to account for temporal trends. 

It would be my personal preference to have equations 6 & 9 listed along with equations 1-3 in the main text, since they are referred to throughout the results and tables.

Reviewer #2: The objectives of the study are clearly articulated.

The methodology used is appropriate to address the stated objectives.

A robust statistical analysis was used to support conclusions.

There are no concerns about ethical or regulatory requirements being met.

Reviewer #3: -Are the objectives of the study clearly articulated with a clear testable hypothesis stated?

Yes.

-Is the study design appropriate to address the stated objectives?

Yes.

-Is the population clearly described and appropriate for the hypothesis being tested?

Yes.

-Is the sample size sufficient to ensure adequate power to address the hypothesis being tested?

Yes.

-Were correct statistical analysis used to support conclusions?

Yes.

-Are there concerns about ethical or regulatory requirements being met?

Yes.

**Results**

-Does the analysis presented match the analysis plan?

-Are the results clearly and completely presented?

-Are the figures (Tables, Images) of sufficient quality for clarity?

Reviewer #1: The numbers reported in the results feel abstract and it is hard to see whether there are meaningful (vs. statistically significant) differences. Is it possible to give greater interpretation of the results? For example, from the DID estimator can you report the OR and describe it in terms of increase in the dependent variable for each additional year that MDA was distributed during the 5 year period?

Reviewer #2: The analysis presented matches the analysis plan.

The results are clearly and completely presented.

I think that the figures (Tables, Images) are of sufficient quality for clarity.

Reviewer #3: -Does the analysis presented match the analysis plan?

Yes.

-Are the results clearly and completely presented?

Yes.

-Are the figures (Tables, Images) of sufficient quality for clarity?

Yes.

**Conclusions**

-Are the conclusions supported by the data presented?

-Are the limitations of analysis clearly described?

-Do the authors discuss how these data can be helpful to advance our understanding of the topic under study?

-Is public health relevance addressed?

Reviewer #1: The authors do a good job clearly stating the study limitations. One important point that is lacking right now is an acknowledgment and discussion around the fact that the LF program was largely designed to sit outside the health system. Community drug distributers are volunteers that are not part of the formal health system. Most LF programs (I cannot personally speak to Bangladesh) do not report LF coverage indicators in routine HMIS platforms. The MDA arm of the LF program is typically quite siloed and separate from routine health services like vaccines, vit A, and MCH. Morbidity services are the exception, as these are often integrated (and least loosely) with central health facilities. This doesn't change the paper, but it is worth explicit discussion, as many readers will be aware of the criticism that the LF program is not 'mainstreamed' into other health services.

Reviewer #2: The conclusions are supported by the data presented.

The limitations of analysis are clearly described.

The authors discuss how these data can be helpful to advance our understanding of the topic under study.

Public health relevance is addressed.

Reviewer #3: -Are the conclusions supported by the data presented?

Yes.

-Are the limitations of analysis clearly described?

Yes.

-Do the authors discuss how these data can be helpful to advance our understanding of the topic under study?

Yes.

-Is public health relevance addressed?

Yes.

**Editorial and Data Presentation Modifications?**

Reviewer #1: I suggest the authors consider the following revisions:

- Line 34: the statement "except 5% of total household expenditures and 20% of total non-food expenditures" is not interpretable in the abstract as written; most readers will have no context for what these percentages mean. Suggest using more general language in the abstract (same note for the author summary)

- Line 37: Is this referring to positive associations between LF program interventions? If so, please specify "LF". 

- Line 103 - 112: is there a reason why DEC is omitted?

- Line 161: this sentence would be easier to interpret if you said, "...was identified by plotting the global positioning system coordinates of the PSU on a 2015..."

- Table 3: Can you add the variable names in parentheses to this table so that it matches what's written in Eqns 1-3? 

- Lines 264 - 276: Greater explanation of subscripts is needed for these equations

- Line 435: Should this instead read, "One possible reason for the increase in FINANCIAL PROTECTION is positive interactions with the health staff..." or something to that effect? This statement seems to be referring to the significant CHE outcome.

Reviewer #2: The manuscript is very well written with very minor revision needed. Please consider the following:

1. Line 33 - consider removing 'the' before incidence.

2. Lines 134 and 135 - consider using abbreviations instead(IEC, MMDP and M&E) instead of writing out in full. Already done in lines 110, 111 and 112.

3. Lines 184 and 185 - consider putting FP after the first family planning and use FP afterwards instead of at the end of the sentence.

4. Line 453 - consider adding 'of' before OOPs.

Reviewer #3: No

**Summary and General Comments**

Reviewer #1: The authors have done a nice job using publicly available datasets to perform a complex analysis to investigate the impact of Bangladesh's LF Elimination Program on tracers of UHC. This paper is well-written and I applaud the authors for their use of well-defined indicators and identifying an analysis that best controls for confounding. As the authors state, this type of analysis is only speculative and therefore cannot be used to establish any causal associations. Nonetheless, I imagine that other country programs or donor institutions may be interested to replicate this type of approach in a different country context, which will be more feasible given the clear guidance provided in this manuscript.

Reviewer #2: The topic is very important for the NTD community as well as for the wider academic world. The members of the NTD community would be very pleased to know that what they do currently has positive beneficial impact on the health systems of endemic countries beside the elimination of the specific diseases. This paper could be a source of motivation for the Donors already involved in funding NTD programmes and can be used as an advocacy tool when targeting new Donors.

I believe that the paper is very well written with very minor revision needed. The issues I have is more with the topic in general than with the paper and the way the paper has been written. 

The topic is very complicated and needs more work for experts to get to a point where there will be a better consensus on the who, what, where, when and how. 

More work has to be done on the following: 

1. Identification and/or recommendation of the best indicators to use. The authors have decided to go with WHO-recommended tracer indicators - CCI and CHE. Are these the best tracer indicators to use for determining the impact of NTD programmes on the health systems or should we consider others to get a more plausible result and conclusion?

2. How do we calculate the indicators we decide to use. I believe that collection of data needed and the calculation of health coverage has many issues we still struggle with both on the part of the health systems involved and the communities that are targeted. Some of these problems will eventually reflect on the results and the conclusions of the paper. I am impressed with the extent to which the authors have gone in calculating CHE. They have used total income, total household expenditure and total household non-food expenditure as denominators in calculating CHE. The use of total income as the only denominator in calculating CHE has been a point of contention for many health economists. I like the fact that the authors have also explored the possibilities when one uses several CHE thresholds in conducting their analysis. The statistical analysis they have use look quite impressive.

3. Identification of data to use for such studies and the credibility of data sources. How credible are coverage data reported in countries for NTDs? DHS and HIES, can they be used for all countries for such studies? I believe that DHS and HIES have their limitations that are reflected in studies when they are used.

4. Equity issues should always be considered but they are still complicated with many unresolved areas. The authors have tried to some extent to use urban versus rural comparisons but LF thrives more in rural areas rather than urban areas which makes the results more difficult to interpret. We need to consider vertical versus horizontal equity because within each of the quintiles we use there could be differences and thus the assumption we make that each quintile we identify is homogenous could be flawed. 

In general, such studies go with so many limitations that the study results become ambiguous and open to conclusions that may not be useful to us as a community. However, this paper is a step in the right direction. The authors have done an excellent job. I hope that the authors here and others will continue to test their methods with other diseases to see how such studies will evolve with time.

Reviewer #3: I highlight the excellent quality of the work that seeks to establish a consistent evaluation process on the implementation impact of the Bangladesh Lymphatic Filariasis Elimination Program, of an intersectoral nature, originating outside the health sector. The challenge of integrating NTD control actions into national health systems has been the major obstacle to more consistent advances in real-world settings. This is an aspect that should be better discussed given the nature of the control program under evaluation, particularly in terms of its sustainability in different contexts and settings, including politico-institutional ones.

Some aspects that could be better discussed, including: the selection process of the indicators under evaluation under a critical-reflexive perspective, the consistency and completeness of the databases used for the analyses (including social determinants), the necessary interface with local health systems in differential (not equitable) performance of rural and urban territories, the classic discussion between health care coverage and its quality. I emphasize the clear approach to limitations of the study enunciated by the authors.

PLOS authors have the option to publish the peer review history of their article (what does this mean?). If published, this will include your full peer review and any attached files.

Reviewer #1: No

Reviewer #2: No

Reviewer #3: No

Figure Files:

Data Requirements:

Reproducibility:

References

---

## [Decision Letter · Decision Letter 1]

10 Oct 2021

Dear Dr. Koporc,

We are pleased to inform you that your manuscript 'Assessing the effects of disease-specific programs on health systems: An analysis of the Bangladesh Lymphatic Filariasis Elimination Program’s impacts on health service coverage and catastrophic health expenditure' has been provisionally accepted for publication in PLOS Neglected Tropical Diseases.

Best regards,

Alberto Novaes Ramos Jr

Associate Editor

Francesca Tamarozzi

Deputy Editor

Reviewer's Responses to Questions

**Key Review Criteria Required for Acceptance?**

**Methods**

-Are the objectives of the study clearly articulated with a clear testable hypothesis stated?

-Is the study design appropriate to address the stated objectives?

-Is the population clearly described and appropriate for the hypothesis being tested?

-Is the sample size sufficient to ensure adequate power to address the hypothesis being tested?

-Were correct statistical analysis used to support conclusions?

-Are there concerns about ethical or regulatory requirements being met?

Reviewer #1: My concerns have been addressed.

Reviewer #2: All previous reviewer comments have been addressed by authors.

Reviewer #3: I have no additional comment or correction to make.

**Results**

-Does the analysis presented match the analysis plan?

-Are the results clearly and completely presented?

-Are the figures (Tables, Images) of sufficient quality for clarity?

Reviewer #1: My concerns have been addressed.

Reviewer #2: All previous reviewer comments have been addressed by authors.

Reviewer #3: I have no additional comment or correction to make.

**Conclusions**

-Are the conclusions supported by the data presented?

-Are the limitations of analysis clearly described?

-Do the authors discuss how these data can be helpful to advance our understanding of the topic under study?

-Is public health relevance addressed?

Reviewer #1: My concerns have been addressed.

Reviewer #2: All previous reviewer comments have been addressed by authors.

Reviewer #3: I have no additional comment or correction to make.

**Editorial and Data Presentation Modifications?**

Reviewer #1: My concerns have been addressed.

Reviewer #2: All previous reviewer comments have been addressed by authors. The manuscript can now be accepted.

Reviewer #3: Accept

**Summary and General Comments**

Reviewer #1: I appreciate the authors' willingness to adapt the manuscript, no further concerns.

Reviewer #2: All previous reviewer comments have been addressed by authors. The manuscript can now be accepted.

Reviewer #3: The most critical considerations and suggestions inserted in my review have been addressed in the current version of the manuscript

PLOS authors have the option to publish the peer review history of their article (what does this mean?). If published, this will include your full peer review and any attached files.

Reviewer #1: No

Reviewer #2: No

Reviewer #3: No

---

## [Editor Report · Acceptance letter]

18 Nov 2021

Dear Dr. Koporc,

We are delighted to inform you that your manuscript, "Assessing the effects of disease-specific programs on health systems: An analysis of the Bangladesh Lymphatic Filariasis Elimination Program’s impacts on health service coverage and catastrophic health expenditure," has been formally accepted for publication in PLOS Neglected Tropical Diseases.

Best regards,

Shaden Kamhawi

co-Editor-in-Chief

Paul Brindley

co-Editor-in-Chief
